# Neural Inverse Source Problems

**Youngsun Wi**      **Jayjun Lee**      **Miquel Oller**      **Nima Fazeli**
Robotics Department, University of Michigan
{yswi, jayjun, oller, nfz}@umich.edu
https://www.mmintlab.com/nisp

**Abstract:** Reconstructing unknown external source functions is an important perception capability for a large range of robotics domains including manipulation, aerial, and underwater robotics. In this work, we propose a Physics-Informed Neural Network (PINN [1]) based approach for solving the inverse source problems in robotics, jointly identifying unknown source functions and the complete state of a system given partial and noisy observations. Our approach demonstrates several advantages over prior works (Finite Element Methods (FEM) and data-driven approaches): it offers flexibility in integrating diverse constraints and boundary conditions; eliminates the need for complex discretizations (e.g., meshing); easily accommodates gradients from real measurements; and does not limit performance based on the diversity and quality of training data. We validate our method across three simulation and real-world scenarios involving up to 4th order partial differential equations (PDEs), constraints such as Signorini and Dirichlet, and various regression losses including Chamfer distance and L2 norm.

**Keywords:** Inverse source problem, Physics informed neural network

## 1 Introduction

We are interested in differential equations $g$ with an unknown external source $f$:

$$g(\boldsymbol{x}, \Phi, \nabla_{\boldsymbol{x}}\Phi, \nabla_{\boldsymbol{x}}^2\Phi, ...) = f(\boldsymbol{x}), \quad \Phi : \boldsymbol{x} \longmapsto \Phi(\boldsymbol{x}). \tag{1}$$

Our objective is to determine the external source function $f(\boldsymbol{x}) \in \mathbb{R}^q$ and the full mapping from spatial/temporal coordinates $\boldsymbol{x} \in \mathbb{R}^s$ to the quantity of interest $\Phi(\boldsymbol{x}) \in \mathbb{R}^r$ from known governing equations $g$ and partial/noisy observations. This problem, known as an inverse source problem [2, 3, 4], is of significant importance in various scientific and engineering domains such as signal processing [5], fluid dynamics [6, 7], optics [8], and more.

To address these inverse source problems in robotics, a common strategy is to use finite element analysis integrated with optimization techniques [9, 10] or physics engine [11, 12]. While these approaches focus on computation efficiencies, they suffer two notable limitations. First, they rely on complex discretization and/or meshing, which affects precision and introduces significant complexity. Second, these approaches require unique treatment for different equations and constraints, which significantly limits their generality. Recent progress in representation learning addresses these issues via learning priors over the simulated source functions, where the data implicitly contains all the constraints and boundary conditions. For example, prior works in manipulation simulate extrinsic contact, a source function exerting forces on deformable [13, 14, 15, 16] or rigid objects [17, 18]. Another example involves simulating sensor deformations as a source function transducing electric signals [19]. However, the performance of these approaches relies heavily on the diversity and quality of training data and does not guarantee adherence to the physics, especially with out-of-distribution inputs. Latent force models [20, 21] present an alternative approach using Gaussian processes to identify unknowns in differential equations; however, they are restricted to linear models, and it remains unclear how complex constraints could be effectively integrated.

In this paper, we propose a method for solving the inverse source problem based on Physics-Informed Neural Networks [1] that enables the simultaneous inference of both the source function

and the full state given partial and noisy observations. Our method offers several advantages over prior work in robotics: compatibility with various nth-order differential equations (including ordinary and partial differential equations), flexibility in integrating various constraints, initial values, and boundary conditions, elimination of the need for non-trivial discretizations like meshing, ease of accommodating gradients from real measurements, and does not limit performance based on the diversity training data. We open-source our codes and a real-world datasets at **URL**.

## 2   Related Works

**Physics Informed Machine Learning:** Physics-Informed Neural Networks (PINNs) have shown promising results in solving differential equations, including ODEs, PDEs, and IDEs. PINN [1] was initially limited to solving 1st or 2nd order differential equations [22, 23, 24]. Recently, [22, 25] succeeded in solving the forward problem of 4th-order spatio-temporal differential equations by incorporating residual layers [22, 23]. Our network architecture builds upon [25, 22]'s work for its capability in solving complex higher-order differential equations. In contrast, our goal is to recover the source and unobserved system states from partial observations and handle constraints.

**Inverse Problems in PINN:** Inverse problems in PINNs primarily focus on system parameter or coefficient identification in sparse and noisy data regimes [26, 27, 28, 29, 1, 30, 31, 32, 33]. Parameter identification is fundamentally a different problem from source function identification; it is about the internal properties and inherent dynamics of the system, whereas we are interested in external influences that drive the system, affecting the transient and steady-state behaviors. Recently, a thermonuclear study [30] showed promising results in identifying the forcing function of a simple first-order spatio-temporal differential equation without any constraints. To the best of our knowledge, no existing work in PINN has focused on inverse source problems with constraints, validation on high-order differential equations, and applications to robotics.

**Inverse Source Problem in Robotics:** External source identifications have seen considerable attention in the robotics from both first-principles and learning perspectives. In addition to the work cited in the Introduction, we refer the reader to Appendix. A.1.

## 3   Methodology

**3.1. Problem Formulation:** Our goal is to learn a mapping $\Psi : \boldsymbol{x} \longmapsto (\Phi(\boldsymbol{x}), f(\boldsymbol{x}))$ that satisfies:

$$g(\boldsymbol{x}, \Phi, \nabla_{\boldsymbol{x}}\Phi, \nabla_{\boldsymbol{x}}^2\Phi, \ldots) = f(\boldsymbol{x}), \quad \forall \boldsymbol{x} \in \Omega_n \tag{2}$$

$$\text{subject to} \quad \mathcal{C}_m(\boldsymbol{x}, \Phi_\theta, \nabla_{\boldsymbol{x}}\Phi, \nabla_{\boldsymbol{x}}^2\Phi, \ldots, f(\boldsymbol{x})) = 0, \quad \forall \boldsymbol{x} \in \Omega_m, \tag{3}$$

where $\boldsymbol{x}$ is a spatial/temporal coordinates we have partial/noisy access to, $\Phi(\boldsymbol{x})$ is the fully/partialy observable quantity of interest(e.g., deformation field, electric field, etc.), $f(\boldsymbol{x})$ is a Lipschitz continuous source function, and $\Omega_n, \Omega_m$ are subsets of a bounded domain $\Omega$. When we approximate the mapping with a neural network as $\Psi_\theta(\boldsymbol{x})$ and $f_\theta(\boldsymbol{x})$, this problem becomes solving for the network parameters $\theta$, satisfying the differential equations $g(\boldsymbol{x}, \Phi, \nabla_{\boldsymbol{x}}\Phi_\theta, \nabla_{\boldsymbol{x}}^2\Phi_\theta, \ldots) - f_\theta(\boldsymbol{x}) = 0$ and $M$ constraints $\{\mathcal{C}_m(\boldsymbol{x}, \Phi_\theta, \nabla_{\boldsymbol{x}}\Phi_\theta, \nabla_{\boldsymbol{x}}^2\Phi_\theta, \ldots, f_\theta(\boldsymbol{x})\}_{m=1}^M$. We cast this problem into a loss:

$$\frac{1}{|\Omega_n|} \int_{\Omega_n} (g_n(\boldsymbol{x}, \Phi_\theta, \nabla_{\boldsymbol{x}}\Phi_\theta, \ldots) - f_\theta(\boldsymbol{x}))^2 \, d\boldsymbol{x} + \sum_{m=1}^M \frac{1}{|\Omega_m|} \int_{\Omega_m} \mathcal{C}_m(\boldsymbol{x}, \Phi_\theta, \nabla_{\boldsymbol{x}}\Phi_\theta, \ldots, f_\theta(\boldsymbol{x}))^2 \, d\boldsymbol{x},$$
$$\tag{4}$$

where the first term represents residual losses $(\mathcal{L}_r)$ of the differential equation (DE), and the second term represents the constraint loss. The constraint loss includes various forms of regression losses $(\mathcal{L}_{reg})$ enforcing desired output at specific coordinates (e.g., Chamfer Distance, Mean Square Error), boundary conditions (e.g., Dirichlet, Neumann, Signorini), and equality and inequality constraints.

**3.2. Modified Multi-layer Perceptron:** We use a modified multi-layer perceptron (MLP), which has been shown to excel at learning PINNs [22, 23]. The modified MLP's $l$-th layer has 2 steps:

$$\boldsymbol{z}'^{(l)} = \tanh(\mathbf{W}^{(l)}\boldsymbol{z}^{(l-1)} + \boldsymbol{b}^{(l)}), \quad \boldsymbol{z}^{(l)} = \boldsymbol{c} \cdot \boldsymbol{z}'^{(l)} + \boldsymbol{d} \cdot (1 - \boldsymbol{z}'^{(l)}). \tag{5}$$

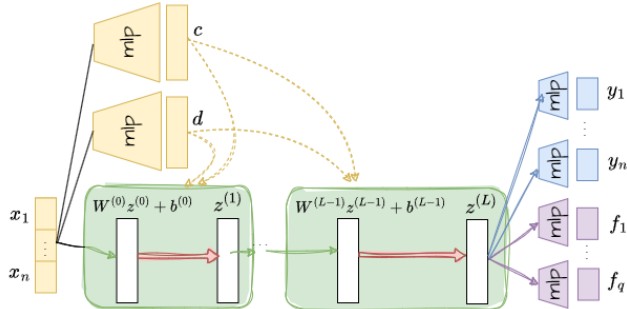

Figure 1: Modified MLP with $L$ layers. Green arrows indicate fully connected layers, and red arrows indicate operations between the fully connected layer output, $\boldsymbol{a}$, and $\boldsymbol{b}$. $L$ is the number of the green box module (Eq. 5) from the input $x$ to the output $\Psi_\theta(\boldsymbol{x}) = (\boldsymbol{y}(\boldsymbol{x}), \boldsymbol{f}(\boldsymbol{x}))$.

The first step is a fully connected layer with weight $\mathbf{W}^{(l)}$, bias $\boldsymbol{b}^{(l)}$, and a $\tanh$ activation function acting on the previous layer output $\boldsymbol{z}^{(l-1)}$. The second step weights the output $\boldsymbol{z}'^{(l)}$ with $\boldsymbol{c}$ and $\boldsymbol{d}$ from separate MLPs with a $\tanh$ activation function, as in Fig. 1. Here, we propose the last fully connected layer unique to each dimension of $y$ and $f$. Please read Appendix. E.1 and E.4 for details.

**3.3. Incorporating Inductive Bias into Network Design:** Many practical robotics systems have low-frequency signals, and building a network with bandwidth matching the underlying signal is essential for robustness to signal noise and successful training. We found that removing the input mappings helps the model be robust to measurement noise (Appendix-E.2), whereas prior works in AI4science [25, 22] highlight the necessity of the input mappings, particularly when focusing on high-frequency signals.

**3.4. Non-dimensionalization:** Ensuring inputs and outputs have reasonable scales is crucial for successful training [25, 34, 35]. We normalize inputs to be within a unit cube $[-1, 1]^s$ and each dimension of the outputs to have similar scales. For instance, if we scale inputs in $[m]$ unit by some value $k$, we scale Young's modulus $[N/m^2]$ by $\frac{1}{k^2}$ and depth measurements in $[m]$ unit by $k$.

**3.5. Loss Weighting:** We found that an effective training strategy is to quickly regress to the partial measurements and then gradually solve for the forcing function. One way to control the convergence speed of each loss term is by updating the loss weights every hundred epochs using each loss term's gradients [25]; however, this approach is expensive in time and space, especially with highly nonlinear and high-order residual losses. Instead, we recommend fixed loss weightings satisfying $50\lambda_r \|\nabla_\theta \mathcal{L}_r\| \leq \lambda_{reg} \|\nabla_\theta \mathcal{L}_{reg}\|$, where $\nabla_\theta \mathcal{L}_{reg}$ and $\nabla_\theta \mathcal{L}_r$ are the gradients of the regression loss and the residual loss. Please see Appendix-E.3 for details.

# 4 Experiments

We present three inverse source problem examples in both simulation and real-world that vary in several key aspects: the order of the differential equations, the degree of observability (ranging from partial to full measurements), and the levels of noise in the data. Additionally, we demonstrate system parameter identification using the same framework in the membrane example.

## 4.1 Double Pendulum Problem

In this example, we validate our method on a well-known non-linear DE system and demonstrate the robustness of our method to different levels of measurement noise. The double pendulum in Fig. 3 is defined over a single variable, time $x$, with the quantities of interest being the cart's displacement $q$, and the angle of `link1` $\psi$ and `link2` $\phi$. The black-box external source function is the force $f$ applied to the cart. The system comprises three differential equations without constraints:

$$x \in \mathbb{R}, \quad \Phi : x \longmapsto \boldsymbol{u}(x) = (q, \psi, \phi) \in \mathbb{R}^3 \tag{6}$$

$$h_1 \ddot{q} + h_2 \ddot{\psi} \cos \psi + h_3 \ddot{\phi} \cos \phi - h_2 \dot{\phi}^2 \sin \psi - h_3 \dot{\psi}^2 = f(x) \in \mathbb{R} \tag{7}$$

$$h_2\ddot{q}\cos\psi + h_4\ddot{\psi} + h_5\ddot{\phi}\cos(\psi-\phi) - h_7\sin\psi + h_5\dot{\phi}^2\sin(\psi-\phi) = 0 \tag{8}$$

$$h_3\ddot{q}\cos\phi + h_5\ddot{\psi}\cos(\psi-\phi) + h_6\ddot{\phi} - h_5\dot{\psi}^2\sin(\psi-\phi) - h_8\sin\phi = 0, \tag{9}$$

where $h_1$ through $h_8$ are scalars determined from Appendix. B.1.

**Dataset:** We construct three datasets without and with observation noise drawn from $\mathcal{N}(0, 0.01)$ and $\mathcal{N}(0, 0.03)$ as visualized in Fig. 3. Each dataset comprises 24 examples of inverted pendulum balancing with various initial conditions (see Appendix. B.2). Each example runs from 0 to 4 seconds at 50 fps, resulting in a total of 200 timestamps $(u_m, \psi_m, \phi_m)_{ts=0:200}$ with the initial condition.

**Training:** When we use the shorthand notations $e_\theta^1$, $e_\theta^2$, and $e_\theta^3$ for the left-hand sides of Eqs. 7, 8, and 9, respectively, the loss function becomes:

$$\arg\min_\theta \frac{\lambda_1}{|\Omega|}\sum_\Omega \left((e_\theta^1 - f_\theta)^2 + (e_\theta^2)^2 + (e_\theta^3)^2\right) + \frac{\lambda_2}{|\Omega_m|}\sum_{\Omega_m}\left((u_\theta - u_m)^2 + (\psi_\theta - \psi_m)^2 + (\phi_\theta - \phi_m)^2\right).$$

Here, the time domain $\Omega$ is $\{x | 0 \le x \le 4\}$, where the domain $\Omega_m \subset \Omega$ is at the observations. The parameters $\lambda_1$ and $\lambda_2$ are weights for the PDE residual loss and the regression loss, respectively.

## 4.2 Softbody Contact Problem

In this example, we demonstrate how our method effectively handles a highly ill-posed inverse problem given partial and noisy observations. This example deals with a softbody system interacting with a rigid terrain with unknown geometry. The goal is to predict 1) a full deformed softbody geometry and 2) contact pressure, given wrist wrench $\boldsymbol{w}_m \in \mathbb{R}^3$ and partial/noisy segmented pointcloud $\mathbf{P}_m$.

A softbody with linear elasticity interacting with a rigid environment is known as a Signorini Problem. This problem is defined in 3D coordinates $\boldsymbol{u}$ of an undeformed softbody $\boldsymbol{x} \in \mathbb{R}^3$. The quantity of interest $\boldsymbol{u}(\boldsymbol{x}) \in \mathbb{R}^3$ is the deformation field, and the forcing function $f(\boldsymbol{x}) \in \mathbb{R}^3$ is contact pressure applied perpendicular to the surface. Following [36], our problem is:

$$\boldsymbol{x} \in \mathbb{R}^3, \qquad \Phi : \boldsymbol{x} \longmapsto \boldsymbol{u}(\boldsymbol{x}) \in \mathbb{R}^3, \qquad \sigma(\boldsymbol{x})\boldsymbol{n} = f(\boldsymbol{x}) \ge 0 \in \mathbb{R} \quad \text{on } \Omega_b, \tag{10}$$

$$div\,\sigma(\boldsymbol{x}) + \rho\boldsymbol{g} = \boldsymbol{0} \text{ on } \Omega, \quad \sigma(\boldsymbol{x}) = \mathbf{D}\varepsilon(\boldsymbol{x}), \quad \varepsilon = \frac{1}{2}(\nabla\boldsymbol{u}(\boldsymbol{x}) + \nabla\boldsymbol{u}(\boldsymbol{x})^T), \tag{11}$$

where $\sigma(\boldsymbol{x})$ is the stress tensor, $\boldsymbol{n}$ is the normal vector at the query, $\rho$ is the density, $\boldsymbol{g}$ is the gravitational constant, $\mathbf{D}$ is the elasticity tensor, and $\varepsilon(\boldsymbol{x})$ is the strain tensor. Domain $\Omega$ is the entire volume of interest with the boundary $\omega_b$. The domain $\Omega_s$ is the four sides of the cubed sponge, where we assume no contact for simplification. Eq. 11 represent the force equilibrium at the contact and the infinitesimal volume at $\boldsymbol{x}$, respectively. $f(\boldsymbol{x})$ is always greater than 0 because normal tractions can only be compressive. Detailed formulations can be found in Appendix-B.3.

**Dataset:** We select 20 examples from [13]'s real-world dataset, where each data sample consists of a single deformed sponge's partial and noisy measurement $(\mathbf{P}_m, \boldsymbol{w}_m)$. The sponge is a 46 mm cube with a Young's modulus $E = 1.1 \times 10^4$ and a Poisson's ratio $\nu = 0.1$. The sponge was attached to the Pandas Franka Emika robot with a force-torque sensor (ATI-Gamma) mounted at the wrist. This dataset includes ground truth contact locations for evaluation, obtained by directly observing the contact location through a transparent acrylic plate.

**Training:** Our model takes the wrist wrench and partial point clouds as input, and it predicts the full deformation and contact pressure. If we define the entire 3D query space as $\Omega = [-0.43, 0.43]^3$, all six surfaces as $\Omega_b$, and four sides as $\Omega_s$, the loss function is defined as:

$$\arg\min_\theta \lambda_1 CD(\mathbf{P}_m, \mathbf{P}_\theta) + \frac{\lambda_2}{|\boldsymbol{w}_m|}\|\boldsymbol{w}_m - \boldsymbol{w}_\theta\|^2 + \frac{\lambda_3}{|\Omega_s|}\sum_{\Omega_s}\|f(\boldsymbol{x})\|^2 + \tag{12}$$

$$\frac{\lambda_4}{|\Omega|}\sum_\Omega \sigma_n(\boldsymbol{x})u(\boldsymbol{x})\cdot\boldsymbol{n} + \frac{\lambda_5}{|\Omega_b|}\sum_{\Omega_b}\|\sigma(\boldsymbol{x})\boldsymbol{n} - f(\boldsymbol{x})\|^2 + \frac{\lambda_6}{|\Omega|}\sum_\Omega\|div\,\sigma(\boldsymbol{x}) + \rho\boldsymbol{g}\|^2 \tag{13}$$

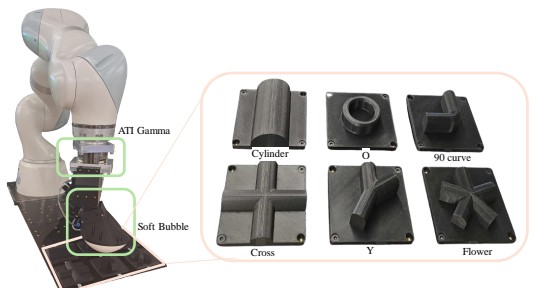

Figure 2: Real-world data collection setup using Soft Bubbles with 6 designed objects.

where $CD$ is a single directional Chamfer Distance from $\mathbf{P}_m$ to estimated deformed surface point-cloud $\mathbf{P}_\theta = \{\boldsymbol{x} + \boldsymbol{u}(\boldsymbol{x}) | \boldsymbol{x} \in \Omega_b\}$ and $\boldsymbol{w}_\theta$ is a wrist wrench estimation calculated from the estimated contact pressure as illustrated in Appendix. C.2. The third term represents an inductive bias that there are no contacts at the sides, and the fourth term represents a Signorini constraint, where $\sigma_n = \boldsymbol{n} \cdot \sigma(\boldsymbol{x}) \boldsymbol{n}$. The last two terms are residual losses from Eq. 11.

### 4.3 Membrane-based Tactile Sensor

Here, we emphasizes the capacity of our method to solve high-order and complex differential equations' inverse source problems. We show our model's ability to perform system identification with the same framework and robustness to partial and noisy observations, unlike the prior works [9, 10].

The classical theory of Von Karman [37] provides a 4th-order PDE model of a thin elastic plate system subjected to large deflections under external forces. This problem is defined in 2D in-plane coordinates $\alpha$ and $\beta$. The quantity of interest is the 3D deformation $[\boldsymbol{u}, w]$ corresponding to the undeformed in-plane coordinates of the membrane. The forcing function $f(\boldsymbol{x}) \in \mathbb{R}$ represents the contact pressure applied perpendicular to the surface normal. The governing equation and the boundary conditions of the membrane sensor are given by:

$$\boldsymbol{x} = (\alpha, \beta) \in \mathbb{R}^2, \ \Phi : \boldsymbol{x} \longmapsto (\boldsymbol{u}, w) \in \mathbb{R}^3, \ D\Delta^2 w - t\frac{\partial}{\partial \beta}(\sigma_{\alpha\beta}(\boldsymbol{x})\frac{\partial w}{\partial \alpha}) - p = f(\boldsymbol{x}) \in \mathbb{R} \quad (14)$$

$$w = 0, \ \boldsymbol{u} = \boldsymbol{0}, \ f(\boldsymbol{x}) = 0 \ \text{on} \ \Omega_b, \quad (15)$$

where $D = Et^3/12(1 - \nu^2)$ is a constant of flexural rigidity consisting of Young's modulus $E$ and Poisson's ratio $\nu$, $t$ is the plate thickness, the biharmonic operator $\Delta^2$ is $\frac{\partial^4}{\partial \alpha^4} + \frac{\partial^4}{\partial \beta^4} + 2\frac{\partial^4}{\partial \alpha^2 \partial \beta^2}$, $\sigma_{\alpha\beta}$ is the shear stress, $\boldsymbol{u} \in \mathbb{R}^2$ represents the in-plane displacement, $w$ denotes the out-of-plane deflection, and $p$ is the traverse load per area from air pressure and gravity. The membrane is subject to a Dirichlet boundary condition in the domain $\Omega_b$. Please refer to Appendix B.4 for further details.

**Dataset:** Our dataset $\mathcal{D} = \{\mathcal{P}_m, p_m\}$ consists of partial and noisy point clouds $\mathcal{P}_i$ and internal air pressures from the pressure sensor $p_m$. We recorded 24 data samples from 6 different geometries with 4 interactions per object, as shown in Fig. 2. For each interaction, the Soft Bubble randomly rotates in yaw $\in [-90°, 90°]$ and lowers until the contact force reaches $8 \sim 11$ N. We use a Kuka arm equipped with an ATI-gamma F/T sensor mounted on the wrist to move the Soft Bubble. The reaction wrench measurement from the F/T sensors is used only for evaluation purposes of our contact pressure estimation. The Soft Bubble's membrane has a thickness of $t = 0.45$ mm and is an ellipse with major axis $a = 0.06$ m and minor axis $b = 0.04$ m.

**System Identification:** Our network can also perform system identifications via optimizing for unknown parameters. When we use an inflated sensor pointcloud without contact, the sourcing term is all zero $f_\theta = 0$ and Young's modulus is the only unknown parameter. The loss function becomes

$$\underset{E,\theta}{\arg\min} \ \lambda_1 CD(\mathbf{P}_m, \mathbf{P}_\theta) + \frac{\lambda_2}{|\Omega|} \sum_\Omega (g_{E,\theta})^2 + \frac{\lambda_3}{|\Omega_b|} \sum_{\Omega_b} (w_\theta^2 + \|\boldsymbol{u}_\theta\|^2 + f_\theta^2), \quad (16)$$

where $g_{E,\theta}$ is the left hand side of Eq. 14, $\Omega = \{(\alpha, \beta, w) | (\frac{\alpha}{a})^2 + (\frac{\beta}{b})^2 \leq 1\}$, $\Omega_b$ is the boundary of $\Omega$, $CD$ is one-directional Chamfer Distance from noisy partial pointcloud measurement $\mathbf{P}_m$ to our predicted full pointcloud $\mathbf{P}_\theta = (\alpha, \beta, 0) + (\boldsymbol{u}_\theta, w_\theta) | (\alpha, \beta) \in \Omega\}$.

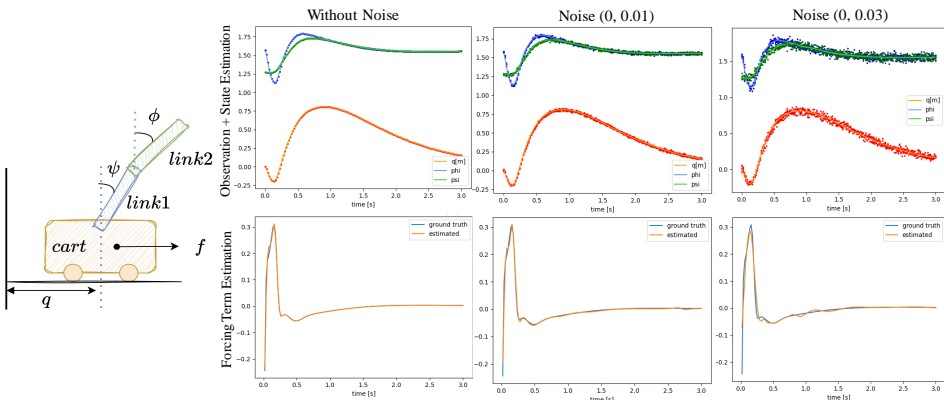

Figure 3: The left panel shows our double pendulum system with positive displacement to the right and positive angles in the counterclockwise direction. In the right panel, we include results without noise and with noise $\sim \mathcal{N}(0, 0.01)$ and $\sim \mathcal{N}(0, 0.03)$ in observations. The scatter plots in the first row depict $q_m$ (red), $\psi_m$ (blue), and $\phi_m$ (green), overlaid with the estimated states $q_\theta$ (orange), $\psi_\theta$ (blue), and $\phi_\theta$ (green) represented by lines. The second row shows the estimated force $f_\theta$ (orange) and the ground truth $f_m$ (blue).

| L1 Error | Full State | | | Source Function |
|---|---|---|---|---|
| | $q$ | $\psi$ | $\phi$ | $f$ |
| No noise | $2.117e\text{-}3$ | $1.587e\text{-}3$ | $7.735e\text{-}4$ | $3.825e\text{-}4$ |
| $\mathcal{N}(0, 0.01)$ | $2.505e\text{-}3$ | $2.808e\text{-}3$ | $1.528e\text{-}3$ | $1.640e\text{-}3$ |
| $\mathcal{N}(0, 0.03)$ | $4.683e\text{-}3$ | $3.945e\text{-}3$ | $5.632e\text{-}3$ | $1.262e\text{-}2$ |

Table 1: The L1 norm error of the double pendulum's full state and source function estimation results with different Gaussian noises. $q$ and $f$ are in meters, and $\psi$ and $\phi$ are in radians.

**Training:** Given a pointcloud under contact $\mathbf{P}_m$, the loss function for contact pressure estimation is

$$\arg\min_\theta \lambda_1 CD(\mathbf{P}_m, \mathbf{P}_\theta) + \frac{\lambda_2}{|\Omega|} \sum_\Omega (g_\theta - f_\theta)^2 + \frac{\lambda_3}{|\Omega_b|} \sum_{\Omega_b} (w_\theta^2 + \|\boldsymbol{u}_\theta\|^2 + f_\theta^2). \tag{17}$$

The only difference between Eq. 16 is non zero $f_\theta$ and that the Young's modulus is not updating.

## 5 Results

### 5.1 Double Pendulum Problem

**Metrics:** We use the L1 error to assess the full states and the sourcing function estimations at unseen time queries with $\times 4$ dense samplings.

**Result:** The results in Tab. 1 and Fig. 3 demonstrate that our model can almost perfectly predict the ground truth for both the full states and the source function when there is no measurement noise. As noise is introduced, our model's accuracy gradually decreases. This trend is more pronounced in the source function estimation because accurately determining n-th order derivatives from noisy data becomes significantly more challenging with noise. Nonetheless, average force errors of $1.262e\text{-}2$ [N] from noise $\mathcal{N}(0, 0.03)$ are still a very small fraction of the scale of the ground truth (ranging from -0.3 to 0.3 [N]) as visualized in Fig. 3.

### 5.2 Softbody Contact Problem

**Metrics:** We consider a point to be in contact when the estimated contact pressure exceeds 1,500 Pa, found to be optimal through grid search. The average Chamfer Distance between the ground truth and the estimated contact patch in our approach is $45 \text{ mm}^2$, whereas the baseline achieves $32 \text{ mm}^2$.

**Baseline:** Our SOTA baseline [13] tackles this ill-posed problem by training a neural network on a dataset comprising a total of 3,000 sponges-environment interactions. One-third of the data includes sponge-box interactions, similar to their real-world dataset's sponges-tables interactions. They utilize Isaac Gym's softbody simulation [38], based on finite element method (FEM), which is non-

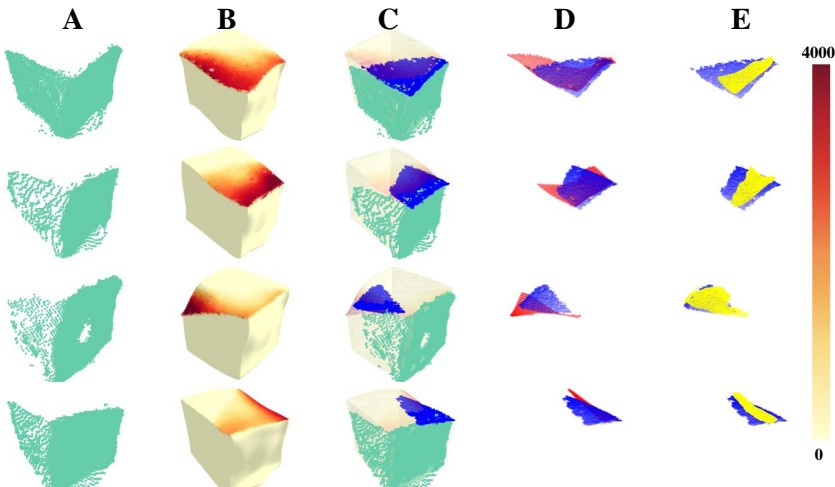

Figure 4: A) Partial pointcloud measurement (green), B) reconstructed deformed geometry where the color represents predicted contact pressure ranging from 0 to 4,000Pa, C) reconstructed deformed geometry overlaid with ground truth contact (blue) and partial observation (green), D) estimated contact location (red) overlaid with the ground truth (blue), E) baseline's contact location (yellow) overlaid with the ground truth (blue).

differentiable and can only address forward problems. The baseline [13] takes wrist wrench, partial point clouds, and a trial code as inputs, and produces signed distances and binary contact as outputs.

**Result:** We demonstrate that our results are both qualitatively and quantitatively comparable to the SOTA baseline [13] (Fig. 4). This is impressive considering that our model had to infer from an infinite number of contact possibilities from partial and noisy measurements and physics equations, whereas the baseline [13] relies on strong priors obtained from the data it was trained on.

### 5.3   Membrane-based Tactile Sensor

**System Identification:** We used $\nu = 0.5$ like typical rubber [9] and $p = 103,320 Pa$ from the pressure sensor measurement. The resulting Young's modulus is $E = 341,260 Pa$.

**Metrics:** We evaluate the predicted contact pressure with 1) contact patch and 2) net contact force estimation. For contact patch evaluations, we use Intersection over Union (IoU) and bidirectional Chamfer Distance (CD). We classify a point in contact if the predicted contact pressure satisfies $f_\theta(\boldsymbol{x}) > \max(2500, \frac{2}{|\Omega|} \int_\Omega f_\theta d|A|)$, where $\frac{1}{|\Omega|} \int_\Omega f_\theta d|A|$ is the average contact pressure across the entire bubble surface area [9]. Here, the ground truth is the intersection of the Soft Bubble pointcloud and the object mesh, using known transformations. For contact force evaluations, we use L2 norm between the estimated wrench at the wrist (Fig. 2) following Appendix. C.2 and the ground truth.

**Baselines:** Our SOTA baseline [9] is a Finite-element (FE) based approach, highly specialized in the Soft Bubble's 3D contact force and contact location estimation. Unlike our approach, [9] requires correspondence tracking between mesh before and after contact, a special system identification process requiring a total of 205 data points, and a result refinement step for handling partial and noisy observations using a convex optimization (CVXPY). We utilized their finest mesh resolution with 749 vertices along with their best-identified model calibration results.

**Results:** Fig. 5 shows examples of bubble-object interactions and the resulting contact pressure estimation from our model. Tab. 2 indicates that our method excels at contact patch predictions for all shapes except for Y, producing 15% higher IoU and 22.2% smaller chamfer distances when compared to the baseline. Fig. 6 shows visualizations of both our and the baseline's contact patch estimations, highlighting that our predicted contact masks are much smoother, less noisy, and more precise at following the shapes of the objects. Although the predicted contact pressures outperformed results on contact patch estimation, the predicted contact force produced a 1.083 N higher contact force L2 error, which is about 10.8% of the average contact force scale, 9.99 N.

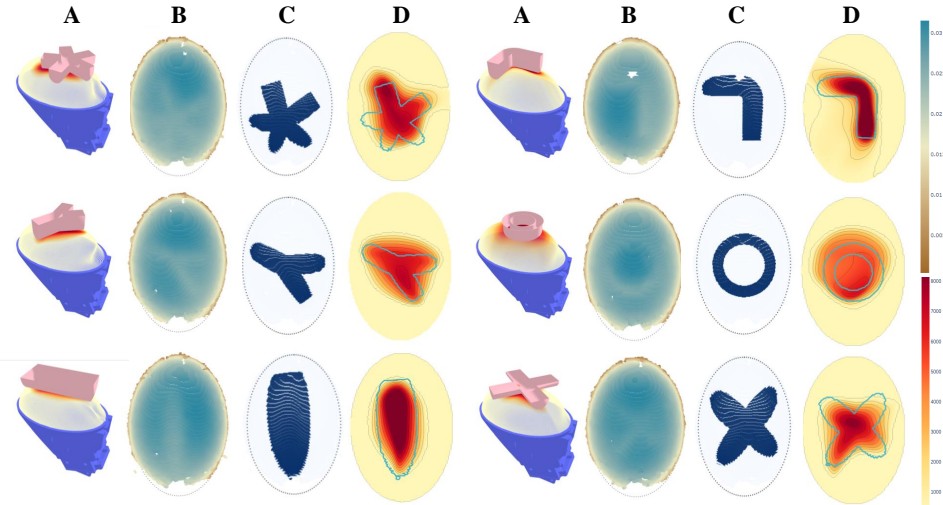

Figure 5: A) Visualization of real-world Soft Bubble and object (pink) interactions. The Soft Bubble is from our model's reconstruction, and the color indicates our model's contact pressure predictions. B) Real-world Soft Bubble point cloud measurement with holes and occlusions. The color represents heights from the x-y plane. The dotted line indicates the actual bubble dimension. C) Ground truth contact mask prediction (navy). D) Estimated contact pressure overlaid with the ground truth contact location indicated in outlines (blue).

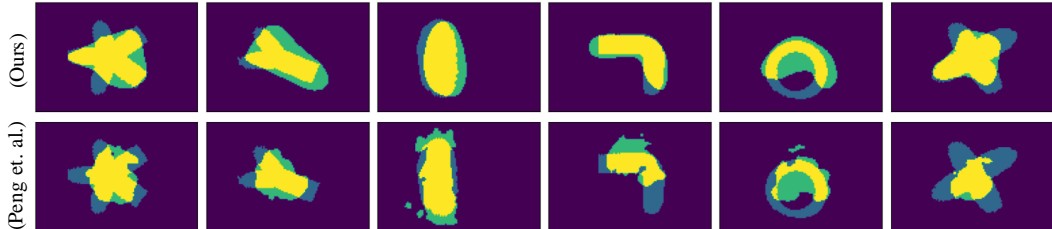

Figure 6: Contact location estimation between ours and SOTA baselines. Yellow region is the overlap between the ground truth and the estimated contact patch, blue is false negative, and green is false positive.

| | Metrics | Object Geometry | | | | | | Average |
| | | Flower | O | Y | Cylinder | Cross | 90 Curve | |
|---|---|---|---|---|---|---|---|---|
| Ours | IoU ↑ | **0.604** | **0.376** | 0.541 | **0.647** | **0.680** | **0.597** | **0.574** |
| | CD ↓ | **1.000** | **2.191** | 1.120 | **0.903** | **0.635** | **0.911** | **1.127** |
| | Force (N) ↓ | 1.900 | 2.056 | 1.741 | 2.758 | 1.350 | 1.514 | 1.886 |
| [9] | IoU ↑ | 0.557 | 0.334 | **0.581** | 0.553 | 0.431 | 0.511 | 0.494 |
| | CD ↓ | 1.026 | 2.357 | **0.933** | 1.083 | 1.750 | 1.514 | 1.444 |
| | Force [N] ↓ | **0.678** | **0.706** | **0.357** | **1.642** | **0.873** | **0.563** | **0.803** |

Table 2: Soft Bubble's contact patch and contact force estimation on six different shapes on our approach and a FEM-based baseline [9]. Force is a L2 norm of the contact force error.

# 6 Future Works and Limitations

While our approach has shown effective in solving inverse source problems in some real-world robotics applications, we have yet to explore real-time inference capacity and representing multiple DE solutions with single networks. Interesting future work is to parameterize multiple DE solutions, similar to methods used in [13, 39, 16], and integrate the latent space inference for potential real-time applications. Additionally, while our method relies on the known governing equations of the system, an exciting direction involves replacing these known differential equations with surrogate models identified from data [40, 41, 42]. This could further enhance the applicability of our method towards unidentified systems.

**Acknowledgments**

This work is supported by the NSF CAREER #2337870 and NRI #2220876 grants. The views expressed here are only of the authors and do not reflect that of the NSF. We thank the members of the Manipulation and Machine Intelligence (MMINT) Lab for their support and feedback. We are also grateful to the reviewers for the excellent comments.

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

| Examples | Initial Values | | | | | |
|---|---|---|---|---|---|---|
| $\psi$ | -0.03 | -0.02 | -0.01 | 0.01 | 0.02 | 0.03 |
| $\phi$ | -0.03 | -0.02 | -0.01 | 0.01 | 0.02 | 0.03 |
| $\dot{\psi}$ | -0.03 | -0.02 | -0.01 | 0.01 | 0.02 | 0.03 |
| $\dot{\phi}$ | -0.03 | -0.02 | -0.01 | 0.01 | 0.02 | 0.03 |

Table 3: We perform 24 examples with offsets for each value of the parameters $\psi, \phi, \dot{\psi}$, and $\dot{\phi}$. In other words, if one parameter has an initial offset, then the offsets for the other parameters were set to zero.

## A    Related Works

### A.1    Unknown External Source Identifications in Robotics

In the field of robotics, there has been a growing focus on identifying unknown sources that influence system behavior. Particularly in the manipulation domain, the system inputs can originate from two primary sources: the robot itself and the environment. Inputs from the robot are typically measurable through various onboard sensors, such as force/torque (F/T) sensors or visio-tactile sensors. In contrast, inputs from the environment are not directly measurable, making their identification a crucial area of research. A significant topic in this research is the inference or identification of contact formations as unknown system inputs. For example, studies such as [13, 14, 16] have focused on identifying contact locations of elastic tools (e.g., sponges, spatulas) using partial point clouds and wrench data. Other studies, like those by [17, 43, 44, 18], have investigated extrinsic contact identification using visio-tactile measurements. In other domains of robotics, unknown system inputs often take the form of external disturbances. For instance, in underwater and aerial robotics, estimating current or wind velocity is essential for achieving optimal robot motion [45, 46].

Additionally, across various robotic applications, identifying the actual input to a sensor (such as force or deformation) from the transduced signals (like electric signals or depth images) is another form of an inverse source problem. For instance, [19] focused on estimating tactile sensor deformations from electric signals, while [9, 10] studied contact pressure estimation using depth camera outputs from sensors. It's important to note that these previous works utilize either data-driven approaches [13, 14, 16, 17, 43, 44, 18, 19] or traditional methods often based on discretizations [9, 10, 45, 46]. These methods have certain limitations, which are discussed in Sec. 1.

## B    Experiment Details

### B.1    Double Pendulum Parameters

The parameters used in Eqs. 7, 8, and 9 are as follows:

$$h_1 = m_1 + m_2 + m_3, h_2 = l_1(m_1 + m_2), h_3 = m_2 l_2, h_4 = l_1^2(m_1 + m_2), \tag{18}$$

$$h_5 = l_1 l_2 m_2, h_6 = l_2^2, h_7 = g(m_1 + m_2), h_8 = m_2 l_2, \tag{19}$$

where $m_1$ is the mass of the cart, $m_2$ is the mass of the first link, $m_3$ is the mass of the second link, $l_1$ is the length of the first link, $l_2$ is the length of the second link, and $g$ is the gravitational constant (9.81 m/s$^2$). The cart's mass is $m_0 = 0.05$ kg, and the lengths of the two links are $l_1 = l_2 = 0.5$ m with masses $m_1 = m_2 = 0.005$ kg.

### B.2    Double Pendulum Initial Values

We give angle offsets in $\psi$ or $\phi$ range from $[-0.3, 0.3]$, and initial angular velocities in $\dot{\psi}$ or $\dot{\phi}$ range from $[-0.3, 0.3]$. Our 24 example's initial values are in Tab. 3.

### B.3 Softbody Contact Problem

Here, we show detailed formulation used for Eq. 13 implementation. First, 3x3 tensors $\sigma(\boldsymbol{x})$ and $\varepsilon(\boldsymbol{x})$ are

$$\sigma = \begin{bmatrix} \sigma_{xx} & \sigma_{xy} & \sigma_{xz} \\ \sigma_{xy} & \sigma_{yy} & \sigma_{yz} \\ \sigma_{xz} & \sigma_{yz} & \sigma_{zz} \end{bmatrix} \text{ and} \tag{20}$$

$$\varepsilon = \begin{bmatrix} \varepsilon_{xx} & \varepsilon_{xy} & \varepsilon_{xz} \\ \varepsilon_{xy} & \varepsilon_{yy} & \varepsilon_{yz} \\ \varepsilon_{xz} & \varepsilon_{yz} & \varepsilon_{zz} \end{bmatrix} = \frac{1}{2}(\nabla \boldsymbol{u} + \nabla \boldsymbol{u}^T) = \begin{bmatrix} \frac{\partial u_x}{\partial x} & \frac{1}{2}\left(\frac{\partial u_x}{\partial y} + \frac{\partial u_y}{\partial x}\right) & \frac{1}{2}\left(\frac{\partial u_x}{\partial z} + \frac{\partial u_z}{\partial x}\right) \\ \frac{1}{2}\left(\frac{\partial u_y}{\partial x} + \frac{\partial u_x}{\partial y}\right) & \frac{\partial u_y}{\partial y} & \frac{1}{2}\left(\frac{\partial u_y}{\partial z} + \frac{\partial u_z}{\partial y}\right) \\ \frac{1}{2}\left(\frac{\partial u_z}{\partial x} + \frac{\partial u_x}{\partial z}\right) & \frac{1}{2}\left(\frac{\partial u_y}{\partial z} + \frac{\partial u_z}{\partial y}\right) & \frac{\partial u_z}{\partial z} \end{bmatrix}. \tag{21}$$

Linear stress and strain tensor relationship $\sigma(\boldsymbol{x}) = \mathbf{D} \cdot \varepsilon(\boldsymbol{x})$ was in practice implemented following Voigt notation for computation efficiency as follows:

$$\begin{bmatrix} \sigma_{xx} \\ \sigma_{yy} \\ \sigma_{zz} \\ \sigma_{xy} \\ \sigma_{xz} \\ \sigma_{yz} \end{bmatrix} = \frac{E}{(1+\nu)(1-2\nu)} \begin{bmatrix} 1-\nu & \nu & \nu & 0 & 0 & 0 \\ \nu & 1-\nu & \nu & 0 & 0 & 0 \\ \nu & \nu & 1-\nu & 0 & 0 & 0 \\ 0 & 0 & 0 & \frac{1-2\nu}{2} & 0 & 0 \\ 0 & 0 & 0 & 0 & \frac{1-2\nu}{2} & 0 \\ 0 & 0 & 0 & 0 & 0 & \frac{1-2\nu}{2} \end{bmatrix} \begin{bmatrix} \epsilon_{xx} \\ \epsilon_{yy} \\ \epsilon_{zz} \\ \epsilon_{xy} \\ \epsilon_{xz} \\ \epsilon_{yz} \end{bmatrix}, \tag{22}$$

Moreover, the divergence is formulated as

$$div\sigma = \nabla \cdot \sigma = \begin{bmatrix} \frac{\partial \sigma_{xx}}{\partial x} + \frac{\partial \sigma_{xy}}{\partial y} + \frac{\partial \sigma_{xz}}{\partial z} \\ \frac{\partial \sigma_{xy}}{\partial x} + \frac{\partial \sigma_{yy}}{\partial y} + \frac{\partial \sigma_{yz}}{\partial z} \\ \frac{\partial \sigma_{xz}}{\partial x} + \frac{\partial \sigma_{yz}}{\partial y} + \frac{\partial \sigma_{zz}}{\partial z} \end{bmatrix}. \tag{23}$$

### B.4 Von Kalman Theory

Following [47, 48], we get the following equations:

$$D\Delta^2 w + N_\alpha \frac{\partial^2 w}{\partial \alpha^2} + 2N_{\alpha\beta} \frac{\partial^2 w}{\partial \alpha \partial \beta} + N_\beta \frac{\partial w^2}{\partial \beta^2} = P + f \tag{24}$$

$$N_\alpha = C(E_{\alpha\alpha} + \nu E_{\beta\beta}), \ N_x = C(E_{\beta\beta} + \nu E_{\alpha\alpha}), \ N_\alpha = C(1-\nu)E_{\alpha\beta}, \tag{25}$$

$$E_{\alpha\alpha} = \frac{\partial \boldsymbol{u}_\alpha}{\partial \alpha} + \frac{1}{2}\left(\frac{\partial w}{\partial \alpha}\right)^2, \ E_{\beta\beta} = \frac{\partial \boldsymbol{u}_\beta}{\partial \beta} + \frac{1}{2}\left(\frac{\partial w}{\partial \beta}\right)^2, \ E_{\alpha\beta} = \frac{1}{2}\left(\frac{\partial \boldsymbol{u}_\alpha}{\partial \alpha}\frac{\partial \boldsymbol{u}_\beta}{\partial \beta} + \frac{\partial w}{\partial \alpha}\frac{\partial w}{\partial \beta}\right), \tag{26}$$

$$D = E\frac{t^3}{12(1-\nu^2)}, \ C = \frac{Et}{1-\nu^2}, \tag{27}$$

### B.5 Design Choices of 3D Printed Objects

The objects we selected mimic real-world items, as shown in Fig. 7. For example, our cylinder resembles cans from the YCB object set (e.g., Pringles can, tuna can). Our dataset was collected in real-world conditions, with 3D-printed shapes specifically designed for precise quantitative analysis of contact location estimations. Using real-world objects (e.g., YCB dataset) for evaluations poses a challenge because obtaining their precise world poses is often difficult. This difficulty stems from object pose tracking methods, such as AprilTags, which are prone to errors caused by camera distortions, depth noise, and occlusions. In contrast, our 3D-printed objects are bolted to the table with known world locations, allowing us to access their precise positions with less than 1 mm of error. In summary, when using general 3D shapes in real-world scenarios, our model will still make predictions, but there will be significant degradation in our evaluation metrics due to the lack of access to ground truth.

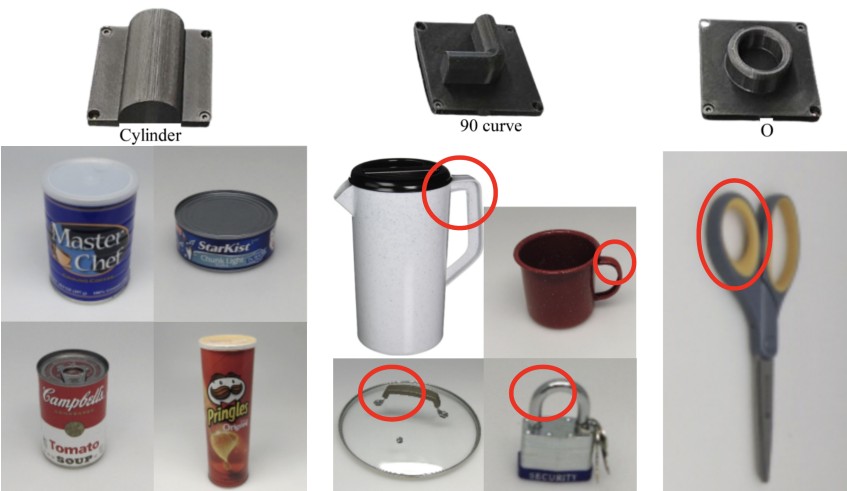

Figure 7: Our 3D-printed objects are analogous to parts of common real-world items. For example, our cylinder resembles the cans from the YCB object set, the 90-degree curve represents the handles on mug and lid as well as locks, and the O-shape is similar to the handle of the YCB scissors.

| Experiment | $s$ | $q + r$ | $L$ | hf | Epochs | Batch size |
|---|---|---|---|---|---|---|
| Double Pendulum | 1 | 4 | 3 | 256 | 100,001 | 201 |
| Softbody | 3 | 4 | 4 | 256 | 8,001 | 5,000 |
| Membrane | 2 | 4 | 4 | 256 | 30,000 | 3,000 |

Table 4: Specifications of training and network hyper-parameters. $s$ is the input dimension, $q + r$ is the network output dimension, $L$ is the number of layers, and hf is the hidden features. For all approaches, we can fit all data points in a single batch.

## C  Result Details

### C.1  Training Details and Hyper Parameters

We train our model on NVIDIA RTX A6000 GPU. For all experiments, we use Adam Optimizer with a learning rate 1e-3 with a decay rate 0.9 per every 5,000 iteration steps. Tab. 4 shows experiment-specific details.

### C.2  Membrane Contact Force Estimation

Given a set of query points $(\alpha, \beta)$, we perform a Delaunay triangulation [49] to generate 2D triangle meshes. Using deformation field predictions from our network, we displace the vertices and calculate the deformed triangle area as described in line 6 of Alg. 1. Then, we calculate the average contact pressure force exerted at each triangle following line 7 of Alg. 1. The total wrench is the summation of the forces from all triangles as described in line 8 of Alg. 1.

### C.3  Membrane Baseline [9] Parameters

For the FEM baseline method [9], we use the same optimized values for the parameters: force penalty $k_{\text{force}} = 0.3322\,\text{N}^{-1}$, displacement penalty $k_{\text{disp}} = 537592\,\text{m}^{-2}$, and membrane-thickness normalized Young's modulus $E = 856\,\text{Pa}$, as reported in [9] and its associated codebase.

---

**Algorithm 1** Contact Force Estimation

---

    **Input:** *queries* - an array of query points $(\alpha, \beta)$ on deflated bubble ellipse (0.06, 0.04)
    **Input:** *deformation* - an array of 3D deformation $[\boldsymbol{u}, w]$ at $(\alpha, \beta)$ from PINN
    **Input:** *pressure* - an array of pressure value $p$ at $(\alpha, \beta)$ from PINN
1: *contact_force* $\leftarrow 0$
2: **for** $\Delta$ in DelaunayTriangulation(*queries*) **do**
3:     $(\alpha, \beta) \leftarrow queries(\Delta)$
4:     $[\boldsymbol{u}, w] \leftarrow deformation(\Delta)$
5:     $v_1, v_2, v_3 \leftarrow [\alpha, \beta, 0] + [\boldsymbol{u}, w]$                    ▷ Deformed vertices of triangle $\Delta$
6:     $area \leftarrow \frac{1}{2}|(v_2 - v_1) \times (v_3 - v_1)|$
7:     force$\leftarrow mean(pressure(\Delta)) \times area$
8:     $contact\_force \leftarrow contact\_force + force$
9: **end for**
10: **return** *force*

---

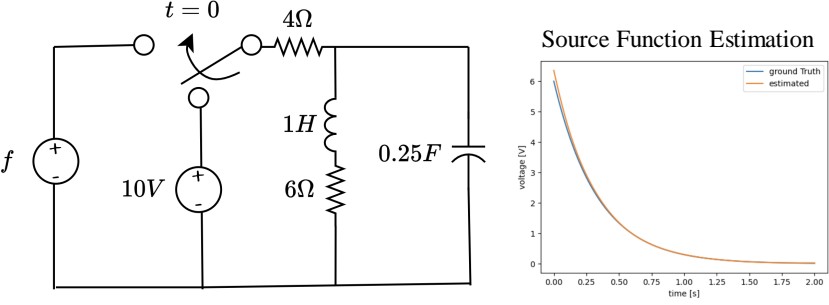

Figure 8: Left panel shows the RLC circuit diagram and the right panel is the source function estimation (orange) overlaid with the ground truth (blue).

## D More Examples

### D.1 RLC Circuit

We show that our approach can be applied even when the right-hand side of the equation consists of the source function and its derivatives:

$$g(\boldsymbol{x}, \Phi, \nabla_{\boldsymbol{x}}\Phi, \nabla_{\boldsymbol{x}}^2\Phi, ...) = g_f(f(\boldsymbol{x}), \nabla f(\boldsymbol{x}), ...), \quad \Phi : \boldsymbol{x} \longmapsto \Phi(\boldsymbol{x}). \tag{28}$$

The RLC circuit in Fig. 8 is described by the equation:

$$\ddot{v} + 7\dot{v} + 10v = \dot{f} + 6f, \tag{29}$$

where $v$ is the voltage at the capacitor and $f$ is the black box source function. For training the dataset, we simulated the RLC circuit with the ground truth source function $6e^{-3t}$, resulting in voltage measurements $v_m = \frac{44}{3}e^{-2t} + \frac{1}{3}e^{-5t} - 9e^{-3t}$. We simulated the circuit for 2 seconds and generated 1k measurements. The source function estimation from training produces only a $5.450 \times 10^{-3}$ V absolute mean error, aligning well with the ground truth as visualized in Fig. 8.

### D.2 Membrane with Noise

In this section, we evaluate our approach and FEM against observation noise, as shown in Fig. 11 and Tab. 5. Specifically, we added Gaussian noise with zero mean and standard deviations of 0.0001, 0.001, 0.002, and 0.003 [m] to depth measurements. Our method demonstrates robustness up to a standard deviation of 0.002 [m], showing only a 12% drop in accuracy, whereas the FEM-based approach and the baseline [9] show performance decreases of 36% and 69%, respectively. This is an impressive result because the baseline [9] explicitly accounts for observation noise via a group

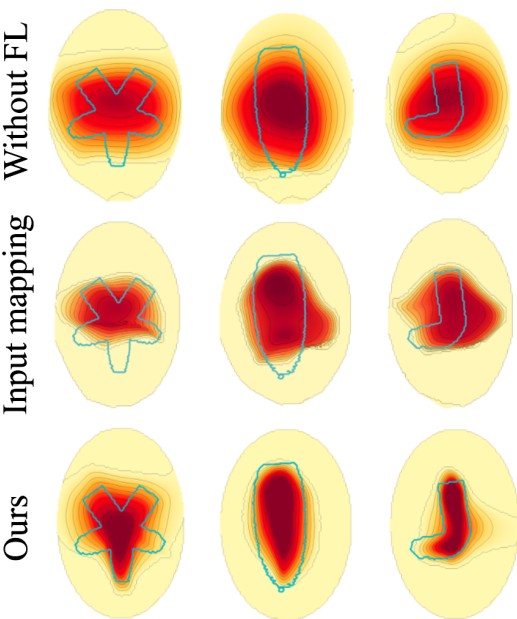

Figure 9: Ablation studies without the final fully-connected layers per each output dimension (Without FL), with sinusoidal input mapping (Input mapping), and our architecture (Ours).

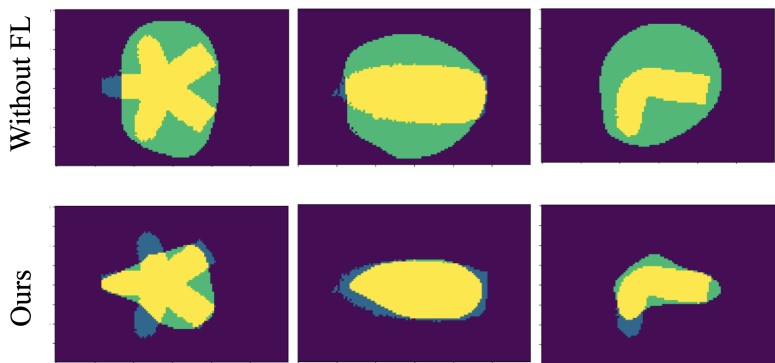

Figure 10: Ablation studies on with (ours) and without the final fully-connected layer for contact patch estimation tasks.

lasso optimization problem, while our approach leverages the inherent noise regularization ability of neural networks without task-specific noise removal treatments.

| Noise | Raw Obs. | 0.0001 | 0.001 | 0.002 | 0.003 |
|---|---|---|---|---|---|
| Ours | **0.710** | **0.703** | **0.696** | **0.624** | **0.240** |
| FEM | 0.252 | 0.251 | 0.208 | 0.160 | 0.120 |
| Baseline [9] | 0.454 | 0.211 | 0.169 | 0.138 | 0.057 |

Table 5: IoU of contact patch detections given raw depth measurements (Raw Obs.) with Gaussian noise having standard deviations of 0.0001, 0.001, 0.002, and 0.003 [m].

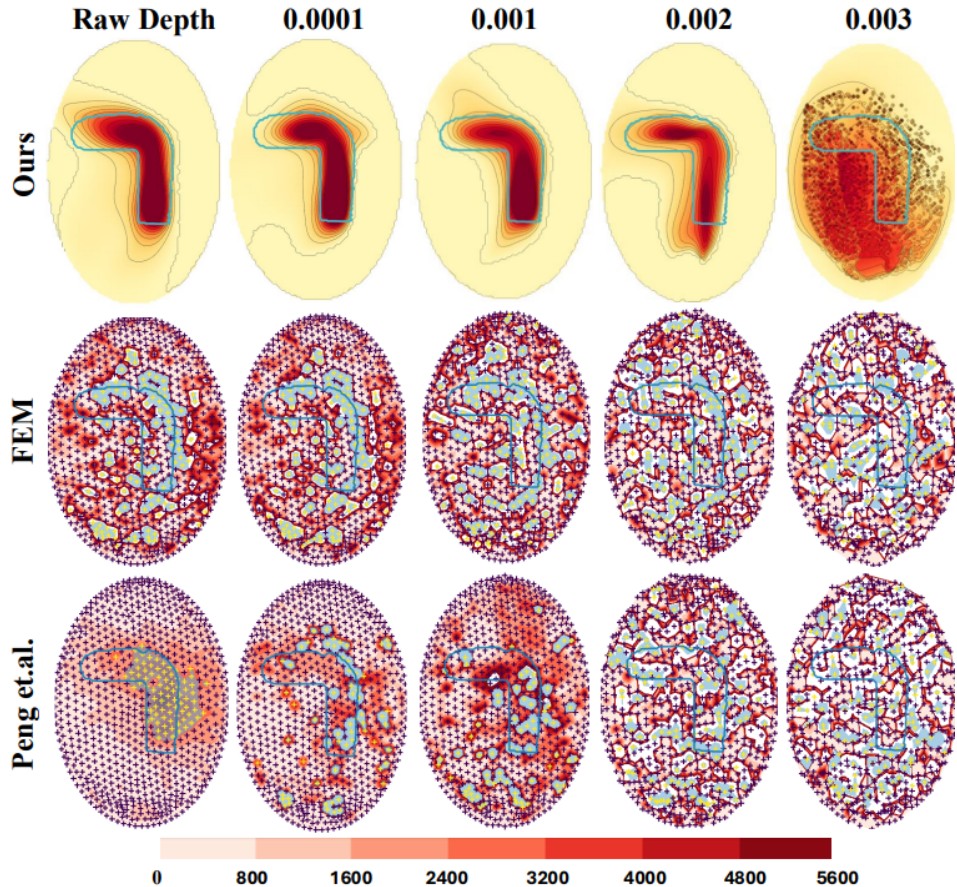

Figure 11: Comparison of contact pressure estimation between our method, FEM, and the baseline [9]. Each column shows the contact pressure distribution given raw depth measurements and with Gaussian noise having standard deviations of 0.0001, 0.001, 0.002, and 0.003 [m]. The baseline includes an internal 'regularization' process to correct point cloud measurement noise. The 'FEM' variant is our modification of [9], which omits the 'regularization' phase and relies solely on FEM for solving the inverse problem. The light blue regions in the FEM and [9] results indicate contact locations found via thresholding.

## E    Ablation Studies

### E.1    Final Layer

We propose to add a fully connected layer for each output dimension, a component absent in the architecture of prior works [25, 22]. We demonstrate that this additional layer brings significant improvements, particularly in source function estimation. Qualitatively, in Fig. 9's contact pressure estimation results, our approach with the final layer shows sharper and more accurate contact patch estimations, closely matching the ground truth. In contrast, the contact pressures estimated by [25, 22] are widely dispersed and fail to capture the shapes of the objects in contact. Furthermore, quantitatively, for three randomly selected examples, we observe an increase in Intersection over Union (IoU) from 0.498 to 0.661 for the flower, from 0.504 to 0.780 for the cylinder, and from 0.298 to 0.667 for the 90-degree curve, respectively, as shown in Fig. 9 and Fig. 10.

### E.2    Sinusoidal Input Mapping

In this section, we demonstrate how the sinusoidal input mapping[25] proposed in prior works makes a model susceptible to measurement noise, which is inevitable in real-world experiments. Using

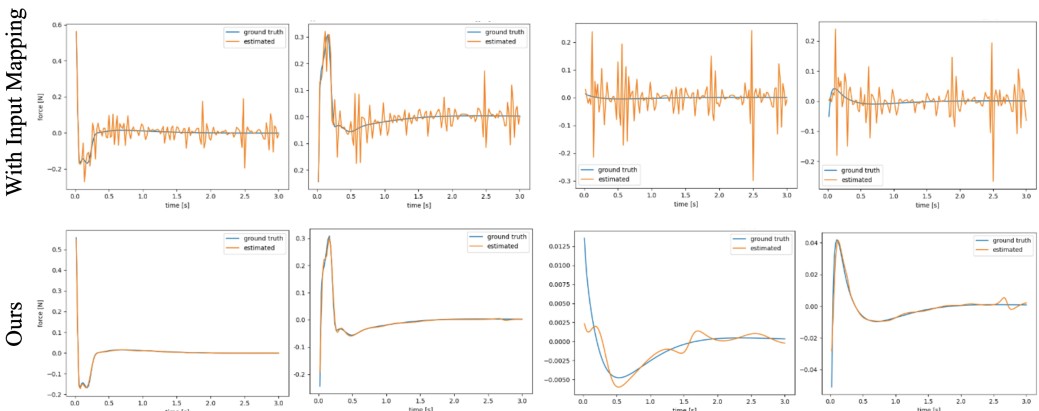

Figure 12: Ablation studies on with sinusoidal input mapping and without the input mapping (ours). Here the noise $\mathcal{N}(0, 1)$ was added to the measurements and the estimated force functions are substantially more noisier than ours without the input embedding.

| L1 | $q$ | $\psi$ | $\phi$ | $f$ |
|---|---|---|---|---|
| Input Mapping | 3.654e-3 | 4.139e-3 | 2.405e-3 | 3.202e-2 |
| Ours | 3.218e-3 | 3.497e-3 | 1.880e-3 | 1.63e-2 |

Table 6: Ablation studies on input mapping given measurement noise $\mathcal{N}(0, 0.01)$. We report the L1 norms of the full state estimation and forcing function estimation. $q$ and $f$ are in meters, and $\psi$ and $\phi$ are in radians.

the same double pendulum system as described in Sec. 4.1 with measurement noise $\mathcal{N}(0, 1)$, we compare the results with and without the sinusoidal input mapping of [25]. We utilize four examples with initial offsets of -0.03 in $\psi, \phi, \dot{\psi}, \dot{\phi}$, respectively. Fig. 12 qualitatively demonstrates that our model without the input embedding exhibits superior robustness in force function estimation under input noise. Quantitatively, Tab. 6 indicates that using input mapping produces 47% more errors in force term estimation compared to not using it (ours).

Moreover, we demonstrate how sinusoidal input mapping degrades the forcing term estimation results in contact location estimation performance. Our proposed architecture, without the input mapping, increased the IoU score from 0.538 to 0.661, 0.606 to 0.780, and 0.399 to 0.667, as visualized in Fig. 9. This represents a performance increase of 22%, 28%, and 67%, respectively.

### E.3  Loss Weighting

Conventionally, as shown in [22, 25], prior works have insisted on keeping the norm of each weighted loss's gradients equal to each other for a fixed number of epochs. However, these weighting methods, which have been classically used in AI4science, are not only 1) expensive in time and space, especially with high-order and non-linear differential equations, but also 2) ineffective for the Inverse Source Problem. This is because the solution to inverse source problems tends to remain near the randomly initialized forcing term while quickly minimizing the DE residual loss. We propose a new approach that involves using a fixed loss term that produces higher loss gradients (by a factor of 50) on regression losses that use partial or noisy observations at the beginning of the training. This approach facilitates the network in first converging to the left side of the DE using observations, so that the forcing term converges to the identified left-side value via the slowly converging DE residual loss. We show the effectiveness of our loss weighting in Fig. 13 and Tab. 7. The results, for instance, show that the loss weighting with $\frac{\lambda_r}{\lambda_{reg}} = 100$ shows 90 % more accuracy

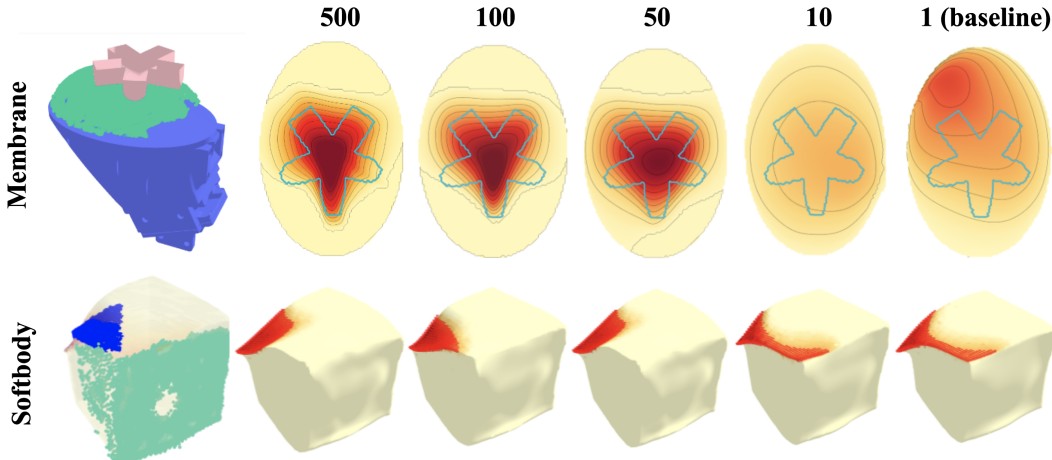

Figure 13: Visualization of the forcing term in the membrane (first row) and soft body (second row) examples with different loss weightings, ranging from $\frac{\lambda_r}{\lambda_{reg}} = 500$ (second column) to 5 (sixth column). The first column of the first row shows the ground truth interaction between the membrane and the 3D-printed objects (pink), with the point cloud observation visualized in green. The second to sixth columns display the contact pressure estimation results, with the ground truth contact patch outlined in blue. The first column of the second row presents the ground truth contact estimation (blue) overlaid with a partially noisy point cloud observation (green). The color scale of the contact pressure distributions matches that of Fig.4 and Fig.5. We observe that contact location detection for both cases significantly deteriorates qualitatively when $\frac{\lambda_r}{\lambda_{reg}} < 50$. A quantitative analysis is provided in Tab. 7.

| $\frac{\lambda_r}{\lambda_{reg}}$ | 500 | 100 | 50 | 10 | 1 (baseline) |
|---|---|---|---|---|---|
| Membrane IoU ↑ | **0.630** | 0.592 | 0.603 | 0.480 | 0.310 |
| Softbody CD ↓ | 46.275 | **25.587** | 14.912 | 55.050 | 55.702 |

Table 7: Quantitative evaluations of results of different loss weighting in Fig. 13, where the evaluation metric for the soft body is IoU and for the membrane is Chamfer Distance (CD).

in IoU for the membrane example and 54 % less errors in softbody example compare to $\frac{\lambda_r}{\lambda_{reg}} = 1$ suggested in the prior works [22, 25].

### E.4    Architecture

We evaluate our method's ability to handle high-order differential equations through the membrane equation's forward problem. This example involves a simplified membrane system described by the equation:

$$D\Delta^2 w = P \tag{30}$$

with Dirichlet boundary conditions. The main differences between Eq. 30 and Eq. 14 are that Eq. 30 does not account for membrane stress in the force balance equation, which holds true only if the deformation is small enough and in-plane deformation $\boldsymbol{u} = \boldsymbol{0}$. Moreover, Eq. 30 assumes there is no contact, i.e., $f = 0$. This simplified system differs from the actual system, resulting in larger deformations. However, removing the stress term makes solution derivation much easier, which is useful for quantitative analysis on deformation predictions. Here, we solve a forward problem given non-updating system parameters $(E, \nu, p, D, t)$ as follows:

$$\arg\min_\theta \sum_\Omega \{D\Delta^2 w_\theta - p\} + \lambda_1 \sum_{\partial\Omega} (\|w_\theta\| + \|\boldsymbol{u}_\theta\|), \tag{31}$$

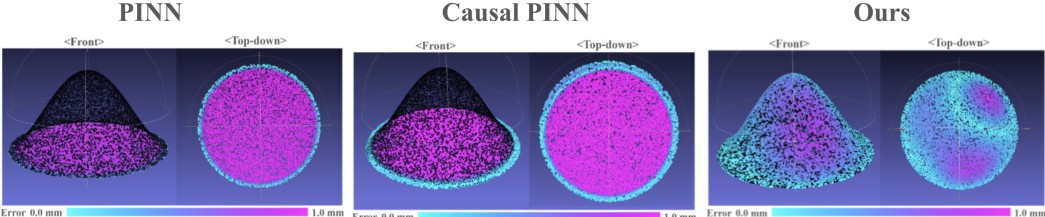

Figure 14: Ablation studies on architectures for solving forward membrane equations. Black pointclouds are the ground truth from the analytical solution, and the colored pointclouds are from each model. The color represents absolute error in deformation predictions.

where $\lambda_1$ is a weighting term. For further simplification, we assume that the closed domain has a fixed radius $r$:

$$\Omega = \{(\alpha, \beta) | \alpha^2 + \beta^2 \leq r^2\}.$$

**Ground Truth:** The solution of the above system is the ground truth vertical deformation $w$ given by $w = \frac{P}{64D}(\alpha^2 + \beta^2 - r^2)^2$.

**System Parameters:** The membrane thickness is $t = 3 \times 10^{-4}[m]$, Young's modulus is $E = 10^9$, Poisson's ratio is $\nu = 0.5$, internal pressure is $P = 0.4\ psi = 2.757 \times 10^3 [N/m^2]$, and $D$ is $Et^3/12(1-\nu^2)$. Also, we assume that the geometry is a sphere for easier ground truth computation with radius $0.04[m]$, which is the length of the bubble sensor's semi-minor axis.

**Evaluation:** We pick optimal network parameters $\theta$ producing the minimal training loss as shown in Eq. 31. We use 10,000 unseen test queries for the evaluations, randomly sampled in the domain. We use $|w_\theta - w|$ for the quantitative evaluations.

**PINN [32]:** The original PINN recommends sampling different query points for every iteration. For every iteration, we use 11,000 query points, where 10k points are from $r = 0.04$ domain and 1,000 points are from the boundary. [32] recommends the following parameters for training: the number of layers is 4, hidden dimension is 20, and activation is tanh. Also, we found these parameters helped convergence: epoch=20k, Adam optimizer with learning rate 1e-3, and loss weighting $\lambda_1 = 1,000$.

**CausalPINN [22]:** CausalPINN recommends the same training data sampling method as the original PINN. CausalPINN has a very similar network architecture to ours except for the input mapping. We follow CausalPINN's 2D Navier-Stokes example's Fourier input mapping $\gamma$ given by:

$$\gamma(\alpha, \beta) = [1, \sin(\alpha k_x w_x), \cos(\alpha k_x w_x), \sin(\beta k_y w_y), \cos(\beta k_y w_y), \tag{32}$$
$$\sin(\alpha k_y w_y)\sin(\beta k_y w_y), \sin(\alpha k_y w_y)\cos(\beta k_y w_y), \tag{33}$$
$$\cos(\alpha k_y w_y)\sin(\beta k_y w_y), \cos(\alpha k_y w_y)\cos(\beta k_y w_y)], \tag{34}$$

where $w_x = w_y = 2\pi$ and $k_x = k_y = 1$. We found the following parameters to be optimal for training: number of layers = 5, hidden dimension = 256, activation = tanh, epoch=20,000, Adam optimizer with learning rate 1e-3, and loss weighting $\lambda_1 = 1,000$.

**Results:** Fig. 14 shows the results of the membrane equation's forward problem with different architectures. The maximum deflection of the ground truth is 36.7 mm, whereas the original PINN's maximum deflection is $\sim 0$ mm, Causal PINN's is 6.5mm, and ours is 36.5mm. Ours only produced 0.2mm of error, whereas the other architectures almost completely failed to solve for the deformations given the air pressure of the membrane system.

