# OpenReview forum: "Neural Inverse Source Problem"
_robot-learning.org/CoRL/2024/Conference — CoRL 2024_

### Official Review · Reviewer_D4sA · 2024-07-20
**Well written paper on solving inverse source problems with PINNs.**

**Originality:** 4
**Technical Quality:** 4
**Clarity Of Presentation:** 4
**Potential Impact:** 3
**Recommendation:** 4
**Confidence:** 4

**Review:**

The paper addresses an important problem in robotics which is the identification of the source function that acts on a PDE or ODE system. Even though the approach seems simple, the simulation and real-world experiments demonstrates the effectiveness of the proposed method. Instead, I would rather say that the simplicity is actually the beauty of the approach. In this way, I think that this is an original idea which can have some impact on the future research on inverse source problems.

Major comments:

- I think that this approach suffers from the main problem of PINNs which is that since the physics are enforced by the loss function evaluated at some points, a non-zero loss means that there are (minor) violations of the physics / constraints at the evaluation points and there is no guarantee what happens in between the points.

- Furthermore, the literature review does not cover all relevant areas as it’s limited to NN approaches. For instance, there is a rich literature on latent force models that address the same issue, see, for instance

Alvarez, Mauricio A., David Luengo, and Neil D. Lawrence. "Linear latent force models using Gaussian processes." IEEE transactions on pattern analysis and machine intelligence 35.11 (2013): 2693-2705.

- Nevertheless, the experimental verification makes a strong case that the presented method can generate reasonable results in the shown use cases

Minor
47: typo “limited to solvinng"

UPDATE AFTER REBUTTAL: Thank you for addressing all my concerns. I updated my score and recommend to accept the paper.

**Quality Of The Limitations Section:**

3

**Questions For Rebuttal:**

-	Can you please extend the literature section including other approaches such as latent force models?

**Robotics Focus:**

4

**Summary Of Paper:**

In this paper, the authors introduce a novel way to solve inverse source problems by exploiting Physics Informed Neural Networks with a modified Loss function. This structure allows to reconstruct the external source function based on partial and noise observations of the system. The proposed method is evaluated on simulations and real-world datasets. In comparison to existing approaches, it shows superior performance regarding the accuracy of the predictions.

**Summary Of Recommendation:**

Simple but effective approach to learn the source function of an ODE or PDE with PINNs with convincing numerical evaluations.

---

### Official Review · Reviewer_dC7P · 2024-07-21
**Introduce a existing tool to robotics area**

**Originality:** 3
**Technical Quality:** 3
**Clarity Of Presentation:** 4
**Potential Impact:** 3
**Recommendation:** 3
**Confidence:** 3

**Review:**

This work uses PINN to solve the inverse problem in the robotics area, e.g. recover the shape from tactile.

The paper demonstrates extensive experiments including simulation and real-world setups.

PINN method is new for the inverse problems in robotics and it may lead to multiple following works.

However, the technique in the paper is not very novel. There are similar PINN network structure designs [1].

I also have some questions about the comparison with FEM results. PINN is known for its slower convergence and low accuracy compared with the FEM method [2]. I am curious about the better performance of PINN in the paper. Is it due to the noisy observation data? Will PINN have better performance for fully observed data?

Also, the scalability is another problem. Is it possible to scale to a real-world dataset but not 3D-printed shapes?

[1] PirateNets: Physics-informed Deep Learning with Residual Adaptive Networks

[2] Can Physics-Informed Neural Networks beat the Finite Element Method?

**Quality Of The Limitations Section:**

3

**Questions For Rebuttal:**

1. Could you explore more on FEM results? Is it due to the noisy data? Can PINN outperform FEM in simulated experiments without noise?

**Robotics Focus:**

4

**Summary Of Paper:**

This paper proposes to use PINN to solve the inverse source problem in robotics

**Summary Of Recommendation:**

The paper introduces a new tool to the inverse problem in robotics. It can have a good impact for the following methods.

---

### Official Review · Reviewer_FvMb · 2024-07-21
**Review of CoRL submission 423**

**Originality:** 2
**Technical Quality:** 3
**Clarity Of Presentation:** 2
**Potential Impact:** 2
**Recommendation:** 2
**Confidence:** 3

**Review:**

The motivation of the paper is sound and clear, and it is important to adopt neural networks to find the solutions of natural dynamics as underlying PDEs in robotics applications. The experiments are extensive and detailed, showing that the PINN-based method can give results for different robotic tasks. However, there are some weaknesses below.

- The core idea is simple and not novel: neural networks are used to parameterize the unknown PDE parts as well as the solution, and the losses are based on PDE residual errors and boundary conditions. The pipeline for training PINNs seems to be typical in the AI4Science area. Therefore, the contribution of introducing PINNs into the robotics area seems to be incremental.

- In different examples, the NN architecture is the same and the only difference lies in different losses and data, which further weakens the contribution and cannot provide insights into how PINNs will be designed differently for different robot tasks.

- The first example is a common non-linear control problem, and it is expected to compare the proposed method with other off-the-shelf control or RL or other learning-based methods, to fairly show the superiority of PINNs through this traditional robot control example.

- The organization of the paper can be improved. The example section is too long and includes some details about the method, experiment setup, and results, which look a bit messy and not coherent from the view of the whole paper.

**Quality Of The Limitations Section:**

2

**Questions For Rebuttal:**

See weakness above.

**Robotics Focus:**

4

**Summary Of Paper:**

This work proposes to solve inverse source problem of PDE for robotics applicaiton, solving source function and state mapping at the same time. The proposed method is based on PINN and many examples are given to show the results, including one real-robot example.

**Summary Of Recommendation:**

Incremental contribution, lack of insight for PINNs in robotics, poor oganization.

---

### Author Rebuttal · Authors · 2024-08-14

Again, we really appreciate your input and feedback. Attached is the updated manuscript for the rebuttal, with the modifications highlighted in blue.

---

### Decision · Program_Chairs · 2024-09-04

**Decision:**

Accept

**Comment:**

**Post-Rebuttal Guidance**
Guidance: ACCEPT. The author rebuttal was very strong, with precise clarifications on what exactly the contributions are, additional comparisons to baselines and ablations. There was 1 strong accept and 1 weak accept. The reviewer (**FvMB**) who provided a weak reject has not acknowledged/responded to the rebuttal nor updated the score - I had posted my thoughts to reconsider their score in light of the rebuttal.

I would like to convey this message to the authors, which was brought up by the reviewer (**dC7P**):
"The authors should also discuss and cite other recent works in the robotics community that have used PINN, for planning [1], and control [2,3].

[1] NTFields: Neural Time Fields for Physics-Informed Robot Motion Planning [2] Neural Optimal Control Using Learned System Dynamics [3] Value Iteration in Continuous Actions, States and Time [4] Physics-Informed Neural Networks for Continuum Robots: Towards Fast Approximation of Static Cosserat Rod Theory"

**Pre-Rebuttal Summary**

Strengths
- Elegant solution
- Strong quantitative evaluation, including on the real world

Weaknesses
- Limited technical novelty
- The presentation of the paper could be more clear
- Questions on generalization to more complex real data (not 3d printed shapes for example)

Dear authors, looks like the reviews are mixed and have several clarifying questions - please take a look.